# Coupling nitrogen removal and watershed management to improve global lake water quality

Xing Yan [1,4], Yongqiu Xia [1,2,4] ✉, Xu Zhao [1,2], Chaopu Ti[1], Longlong Xia [1], Scott X. Chang [3] & Xiaoyuan Yan [1,2] ✉

Lakes play a vital role in nitrogen (N) removal and water quality improvement, yet their efficiency varies due to differing watershed N input and lake characteristics, complicating management efforts. Here we established the N budget for 5768 global lakes using a remote sensing model. We found that watershed N input reduction and lake water quality improvement are non-linearly related and depends on lake N removal efficiency. A 30% reduction in N loading in watersheds with high N removal efficiencies can improve cumulative water quality by over 70%. Stricter reduction could accelerate achieving water quality goal ($\leq 1$ mg N L$^{-1}$), shortening the time by up to 30 years for most lakes. However, heavily polluted lakes with low N removal efficiencies (50 of 534 lakes with >1 mg N L$^{-1}$) may not achieve the UN's clean water SDG by 2030, even with a 100% N input reduction. Our research highlights the need for targeted N management strategies to improve global lake water quality.

Since the United Nations introduced 'clean water and sanitation' as one of the 17 sustainable development goals (SDGs)[1], water quality protection has been one of the global focuses to meeting the SDGs[2–4]. Although many efforts have been made to achieve the different targets in SDGs[5,6], water quality degradation is still a problem worldwide, from the Gulf of Mexico in America[7] to Taihu Lake in China[8] and the Baltic Sea in Europe[9]. Water quality degradation threatens the health of humans and ecosystems, including reductions in biodiversity, increased occurrences of harmful algal blooms, and reductions in drinking water quality[10,11]. Unfortunately, the growing population and climate change are expected to accelerate water quality degradation[12,13], especially in global lake systems, stressing the urgent need for water quality improvement.

Managing soil nitrogen (N) input in watersheds to reduce N loading to downstream lakes is one of the critical strategies to improve lake water quality[2]. Efforts such as reducing fertilizer application, constructing wastewater treatment plants, and planting cover crops have been widely implemented to reduce N releasing to lakes[14–16].

Despite these efforts, the relationship between watershed N management and lake water quality improvement remains unclear, attributable to factors such as the large variation in N release from the land, the legacy of N flows, and the complex biogeochemical N cycles[7,17,18]. Among these reasons, the inadequate consideration of biogeochemical processes involved in lake N removal is one of the key barriers to understanding the relationship between watershed N management and lake water quality improvement[2,15,19]. Lake N removal through denitrification, the dominant process for permanent N removal through converting nitrate (NO$_3^-$) to gaseous N (N$_2$)[20], can substantially remove N from watersheds (e.g., 36 – 47% in the Mississippi River watershed[2,21]). However, the impact of the lake N removal capacity on lake N budget has rarely been evaluated in designing strategies to restore lake water quality. In particular, the long-term dynamic response of water quality improvement to watershed N management has not been adequately considered in studying the relationships among watershed N management, watershed N loading to lakes, N removal from lakes, and lake water quality improvement[22,23]. This may

[1]State Key Laboratory of Soil and Sustainable Agriculture, Changshu National Agro-Ecosystem Observation and Research Station, Institute of Soil Science, Chinese Academy of Sciences, Nanjing, PR China. [2]University of Chinese Academy of Sciences, Nanjing, PR China. [3]Department of Renewable Resources, University of Alberta, Edmonton, Alberta, Canada. [4]These authors contributed equally: Xing Yan, Yongqiu Xia. ✉e-mail: yqxia@issas.ac.cn; yanxy@issas.ac.cn

be attributed to accurately quantifying the varied N removal rates in lakes through denitrification is challenging due to the high background concentrations of atmospheric $N_2$, especially at the landscape scale[24-26].

We recently developed a remote sensing model to estimate lake N removal at the landscape scale by linking several key variables (e.g., concentrations of dissolved carbon, N, and oxygen (DO), and temperature of lake water (WT)) controlling N removal with remote sensing data, e.g., chlorophyll-*a* (Chl*a*), chromophoric dissolved organic matter (CDOM), and WT[27] (see Methods). Using the remote sensing model that we developed, we estimated N removals in 5768 lakes on a global scale. We then developed and validated a dynamic mass balance approach to couple the estimated N removal to the lake N budget. This enabled us to evaluate how lake water quality improves with different watershed N management scenarios. In this study, we try to answer the following three questions: (i) what are the global patterns of N removal in lakes; (ii) how much improvement of lake water quality can potentially be achieved through watershed N management; (iii) what strategies can be employed to achieve global lake water quality goals towards SDG targets?

## Results and Discussion
### Linking lake N loading and concentration with model-estimated lake N removal
By linking the global lake N removal estimated using the remote sensing model to the concentration of N in lake waters and lake N loading (N entering a lake from surrounding surface rivers and underground water transport and direct atmospheric N deposition, see Methods and Supplementary Fig. 1), we show that N removal plays a crucial role in lake N budget and leads to nonlinear relationships between lake N loading and N concentration in lake waters. The anticipated hotspots of lake N loading, N concentration, and N removal rate simultaneously exist in areas with high anthropogenic reactive N input (Fig. 1). Higher N input to watershed soil facilitates N transport into the lake through

runoff, leading to elevated lake water N concentrations and increased N removal capacity[28-30]. However, the global distribution of lake N removal exhibits a distinct pattern of heterogeneity compared to the N loading and N concentration in lake waters, driven by the nonlinear relationships among these extensively interacting items of the N budget. The net anthropogenic N input at the watershed scale (NANI, the main source of lake N loading) is normally distributed (Fig. 1a). However, the lake N removal rate and lake water N concentration follow a skewed normal distribution (Fig. 1b). The NANI in the Midwest US is lower than that in Europe. However, lake N removal rates are comparable between these two regions. Although the Yangtze River basin exhibits substantially higher NANI and lake water N concentrations than the Mississippi watershed, the lake N removal rates do not differ.

Furthermore, by calculating the ratio of annual lake N removal to lake N loading (lake N removal ratio; Fig. 1d), we find that 2049 of the 5768 lakes remove >100% of the loaded N, suggesting that these lakes have a strong self-purification capacity. Previous studies have rarely reported a lake N removal ratio greater than 100%, mainly because lake N removal rates were usually estimated based on the mass balance approach, in other words, based on the difference between N loading and N export from the lake[31]. The N removal ratio of the lake cannot exceed 100% when the N removal rates are estimated using the mass balance approach. However, both the lake N loading from an upstream watershed and the stored N in lakes (the product of lake N concentration and amount of lake water) provide the substrate for the N removal process. This suggests that lake N removal can exceed N loading, as part of the removed N may originate from N already stored in the lake. These lakes with an N removal ratio of >100% are mainly distributed in areas with low watershed N input (e.g., Eastern Europe and Siberia), as the N removal efficiency is higher under low lake N loading[32,33]. We suggest that the reduction of watershed N input may not be required in these lakes to achieve the desired water quality because their water quality has the potential to self-improve due to the strong N removal capacity and low N loading.

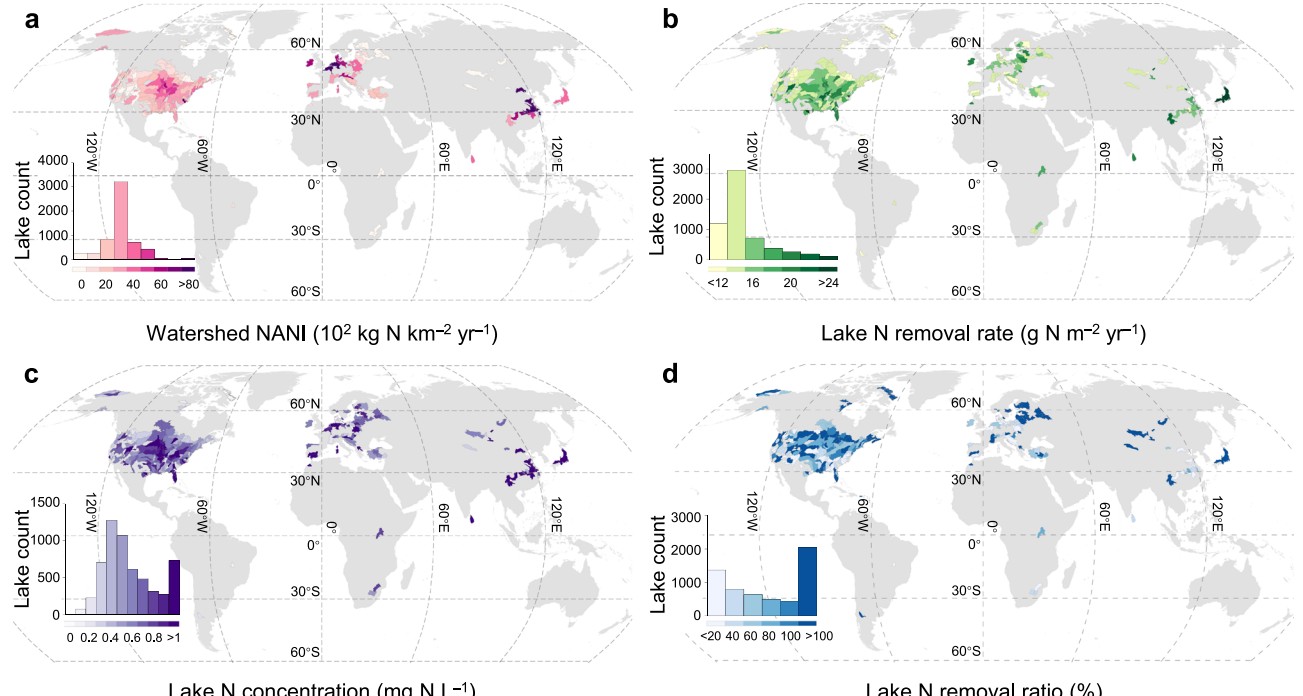

**Fig. 1 | Global patterns of lake N budget and removal. a** Watershed net anthropogenic N input (NANI, the main source of lake N loading). **b** Lake annual N removal rate. **c** Lake water N concentration. **d** Lake N removal ratio (the ratio of annual lake N removal to lake N loading, see Methods). Data are shown at the HydroSHEDS level-5 watershed scale; gray areas indicate areas with no data available.

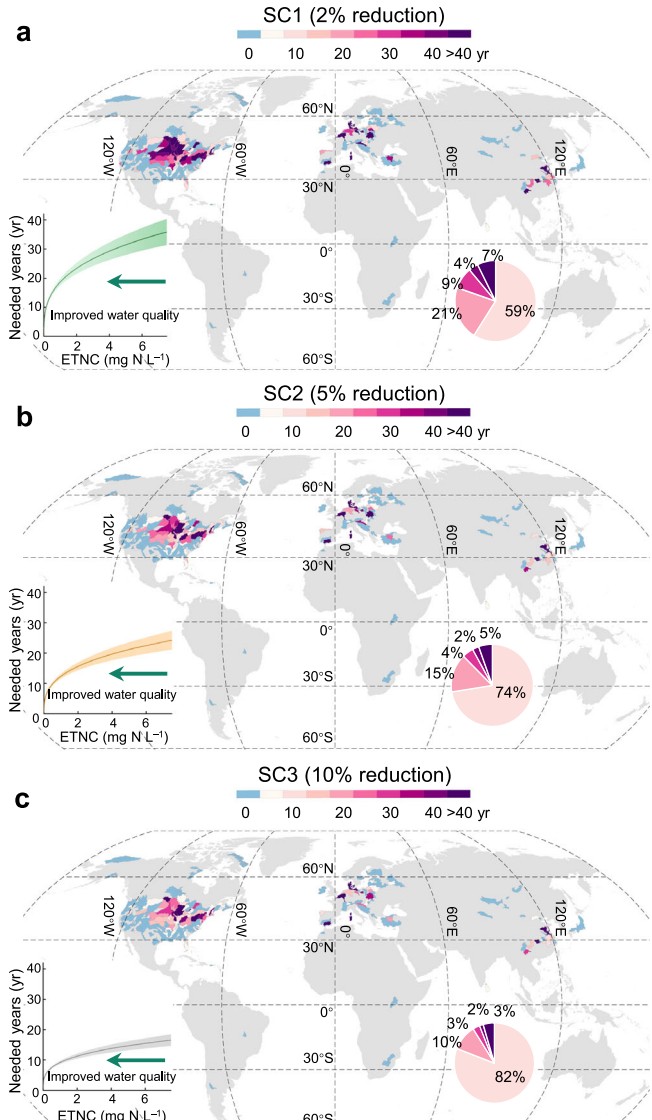

**Fig. 2 | Global patterns of the time needed to achieve the water quality goal (≤1 mg N L⁻¹) under different watershed N input reduction scenarios.** Three scenarios are (**a**) 2% (low-level reduction, SC1), (**b**) 5% (intermediate-level reduction, SC2), and (**c**) 10% (high-level reduction, SC3) reduction in annual watershed N input compared to those in each preceding year. The sub-graphs in the lower left corner show the relationship between the years needed and the exceedance of targeted N concentration in lake waters (the difference between the current concentration and the water quality goal of ≤1 mg N L⁻¹, ETNC). The pie charts represent the proportion of lakes that require 0–10, 10–20, 20–30, 30–40, and >40 years to achieve the water quality goal. Data are shown at the HydroSHEDS level-5 watershed scale. Blue polygons indicate that the lake water quality is ≤1 mg N L⁻¹, and there is no need to reduce watershed N input. Gray areas indicate areas with no data available.

In this study, however, 3719 of the 5768 lakes remove ≤100% of N loading, suggesting that these lakes are a net source of N for downstream water bodies (Fig. 1d); those lakes are mainly located in areas with high watershed N input. The N removal processes in those lakes are characterized by a biological saturation effect under high N levels[32,33]. Although these lakes can remove a large proportion of lake N loading (having a mean global N removal ratio of 37%), it is necessary to reduce N input in these watersheds as the ≤100% N removal ratio indicates that the water quality of these lakes is degraded by N loading from the watersheds.

## Potential improvement of lake water quality

We selected 534 lakes from the database that have N concentration >1 mg N L⁻¹ and N removal ratio ≤100% for analysis (as a case study) based on the mass balance approach (see Methods and Supplementary Fig. 1). We carried out simulations for three scenarios with watershed N input reductions of 2% (low-level reduction, SC1), 5% (intermediate-level reduction, SC2), and 10% (high-level reduction, SC3) of annual watershed N input based on N input data for each preceding year. For each scenario, simulations were run for each lake to estimate the time needed to achieve the water quality goal (1 mg N L⁻¹ of total nitrogen (TN)). A long time is needed to achieve the water quality goal for lakes with high N concentrations, particularly under the SC1 scenario (Fig. 2). Improving water quality becomes increasingly challenging as lake N removal capability decreases with decreasing lake N concentration. Specifically, it takes approximately 5 ± 1 years to reduce the lake N concentration from 3 to 2 mg N L⁻¹, compared to 19 ± 4 years to reduce the lake N concentration from 2 to 1 mg N L⁻¹ under the SC1 scenario. Furthermore, we evaluated the trade-off between the intensity of N input reduction in watersheds and the time required to achieve the water quality goal. Only 59% (51–63%) of the lakes could reach the water quality goal within 10 years under the low-level watershed N input reduction of SC1, while the SC3 scenario could enable 82% (77–85%) of the lakes to reach the water quality goal within the same time frame.

We next set four scenarios with different timeframes (i.e., 10, 20, 30, and 40 years) to achieve the water quality goal of ≤1 mg N L⁻¹ and then evaluated the required reduction intensity in watershed N input. If the water quality goal is to be achieved in 10 years, a reduction of more than 200 kg NANI km⁻² yr⁻¹ (compared with the value in each preceding year) is required for some hotspot watersheds, including the Yangtze River basin and Mississippi watershed (Fig. 3a), to achieve the water quality goal. In addition, 77 of the 534 lakes could not achieve the water quality goal even under a 100% reduction in watershed N input, as the high lake N concentration and low removal ratio necessitate more time to achieve the water quality goal. However, the pressure to reduce watershed N input is greatly reduced under a 40-year timeframe; only a reduction of 50 kg NANI km⁻² yr⁻¹ than that in each preceding year is required to achieve the water quality goal for most of the above hotspot watersheds (51 out of 54) (Fig. 3d). Even so, 29 watersheds still could not achieve the water quality goal under the 40-year timeframe; those watersheds are mainly located in the Upper Mississippi region in the US, the lower reaches of the Huaihe River of China, and some watersheds in Europe.

## Strategies to achieve the clean water target of SDG before 2030

Given that lake N concentration and water quality improvement efficiency vary among lakes[2,4], applying uniform N input reduction strategies yields differing outcomes globally[34,35]. We divided global lakes in our database into three types to explore specific solutions for each lake type to achieve the clean water target before 2030, by which we are supposed to achieve the UN's SDGs (Fig. 4a). Given that 73% of the data in our dataset were collected between 2005 and 2015, with the average being 2008, the analysis to achieve the SDGs by 2030 is aligned with a 20-year time limit scenario in our analysis (see Methods and Supplementary Fig. 2).

Type I lakes (5234 lakes) are those that do not need to have watershed N input reduced to achieve SDG's clean water target. Out of those lakes, 5036 lakes have already met the water quality goal of ≤1 mg N L⁻¹ (with a mean TN concentration of 0.55 mg N L⁻¹), and there is no need to reduce watershed N input beyond the current N input level within these lakes. The other 198 type I lakes do not yet meet the water quality goal, but the capacity for N removal is greater than the N loading rate (with a mean N removal ratio of 218%), and there is also no need to reduce watershed N input for these lakes.

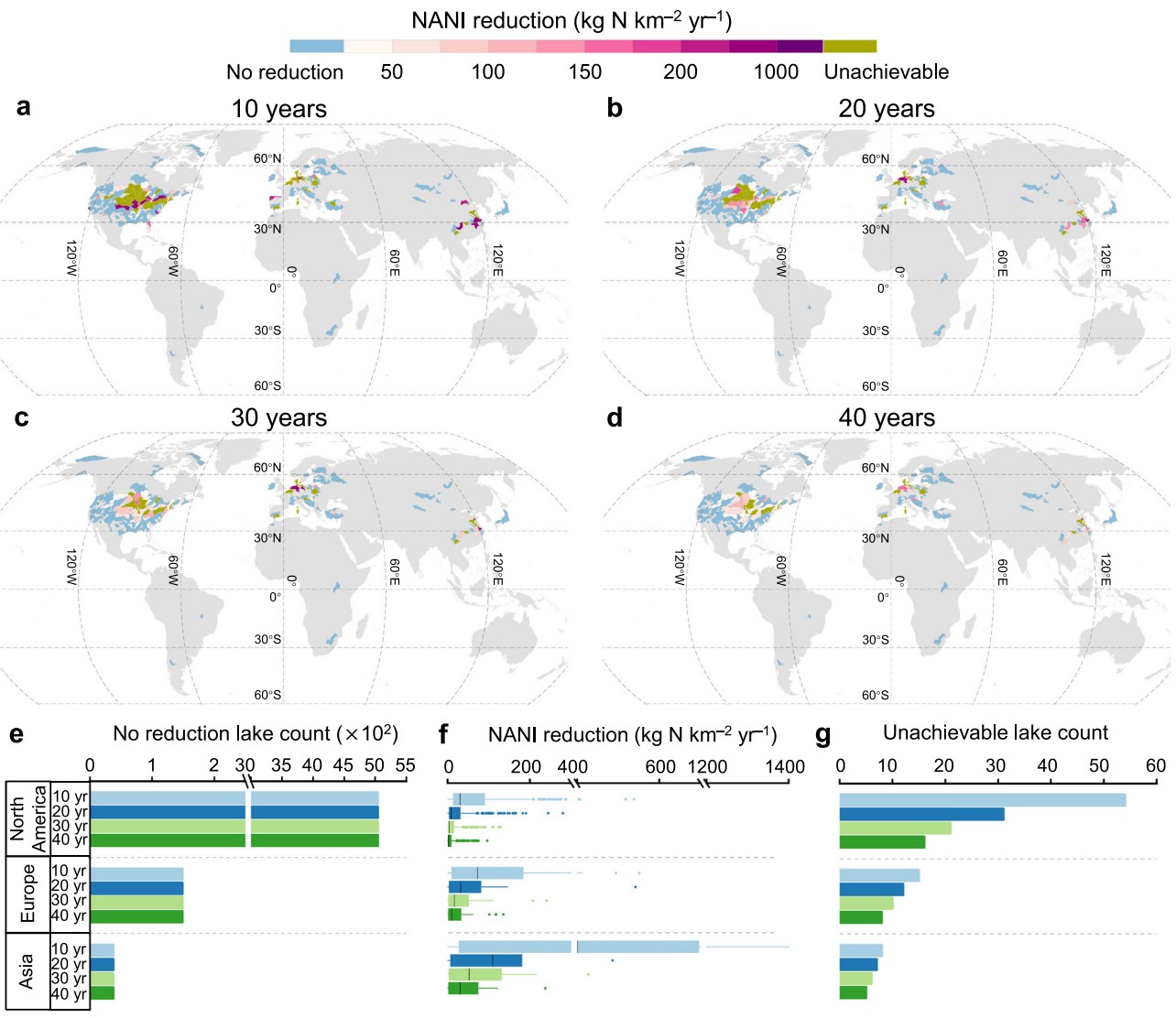

**Fig. 3 | Global patterns of the required reduction in watershed net anthropogenic N input (NANI) to achieve the lake water quality goal (≤1 mg N L⁻¹) within 10, 20, 30, and 40 years. a–d** Blue polygons represent lakes with ≤1 mg N L⁻¹, and for those lakes there is no need to reduce NANI. The deepest yellow polygons represent lakes that could not achieve the water quality goal within the target timeframe, even if the NANI in those watersheds is reduced to zero. Data are shown at the HydroSHEDS level-5 watershed scale. Gray areas indicate areas with no data available. **e** Number of regional lakes that require no reduction in NANI to achieve the water quality goal. **f** Regional NANI reduction required to achieve the water quality goal. **g** Number of regional lakes that cannot achieve the water quality goal within the target years regardless of watershed management practices.

Type II lakes (484 lakes) could achieve the water quality goal within 20 years by reducing an average of 13% lake N loading received from upstream in the watershed. The watershed N management strategies in type II lakes to meet the water quality goal depend on the ratio of reduction in lake N storage to the reduction in lake N loading (water quality improvement efficiency). Lakes with high water quality improvement efficiency have large reductions in lake N storage but small reductions in lake N loading; watershed N management should have higher water quality improvement benefits for those lakes. Lake water quality improvement efficiency increases with the lake N removal ratio under SDG's clean water target before 2030 (Fig. 4b), such as the increasing water quality improvement efficiencies in Poyang Lake, Taihu Lake, and Ge Lake in China, Cayuga and Winnebago Lake in the US, and Lake Sihlsee in Europe (Fig. 4c–h). We find that a 30% reduction in cumulative N loading in target watersheds with high lake water quality improvement efficiencies can achieve more than 70% of the cumulative water quality improvement (sum of reduction in lake N storage for all lakes) (Supplementary Fig. 3).

Type III lakes (50 lakes) are those that could not achieve water quality goals before 2030, even with a 100% reduction (an extreme scenario) in lake N loading received from upstream in the watershed. This may be attributed to the heavy pollution in these lakes and the low water quality improvement efficiency, such as lakes in the Upper Mississippi watershed in the US and in the lower reaches of the Huaihe River in China. In addition to reductions in lake N loading, enhanced removal in lake N storage, including sediment dredging and algae salvaging, are also required to achieve the water quality goal in these lakes[36,37].

Given the challenges faced by policymakers to achieve the SDG goals on a global scale[1,38,39], there is an urgent need to quantify lake N removal efficiencies for formulating strategies to improve water quality in target watersheds. However, each watershed and lake is unique, and there will not be a 'one size fits all' solution for all watersheds. Developing a specific water quality improvement strategy for each lake is often limited by the availability of detailed data on the lake's N budget[40–42]. Our study quantifies the relationship between the amount of the reduction in watershed N input and improved lake

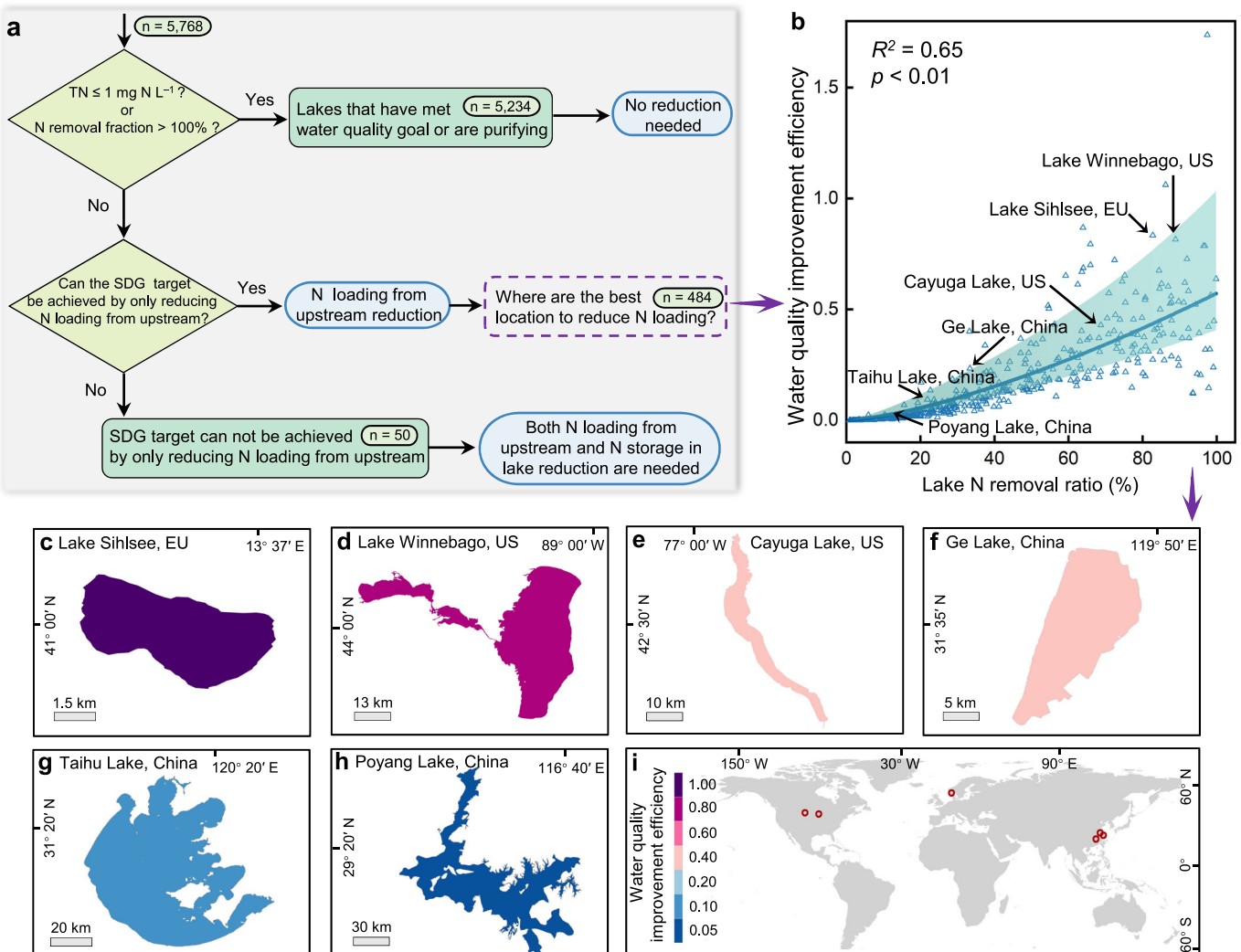

**Fig. 4 | Implementation of strategies to achieve the UN's SDG's clean water target before 2030. a** A flow chart for classifying lakes into three types to design a strategy for achieving the water quality goal for each lake in this dataset. **b** Relationship between lake water quality improvement efficiency and lake N removal ratio. Lakes with high water quality improvement efficiency indicate high water quality improvement but low watershed N input reduction. **c–i** Typical lakes with different water quality improvement efficiencies in this study. Water quality improvement efficiency is dimensionless and calculated as the ratio of the reduction in lake N storage to the reduction in lake N loading.

quality on a global scale. The results suggest the need for target measures to reduce external N loading to certain lakes, and save resources by not focusing on lakes that already meet the water quality criteria or are likely to reach this level with no management measures. Moreover, our study provides a specific strategy for watershed management of 5768 global lakes to achieve the UN's SDG goals for water quality improvement.

## Methods

This section details our analytical methods, the modeling approach, and underlying datasets for estimating global lake N removal, lake N budget, and potential strategies for achieving water quality goals under various watershed N management scenarios.

### N removal and N budget in global lakes

**Overview of the remote sensing model for lake N removal estimation.** The remote sensing model was used to estimate lake N removal[27]. In principle, most of the environmental variables that control lake N removal (e.g., dissolved carbon and N, DO, and WT) can be directly or indirectly derived from remote sensing data (e.g., Chl*a*, CDOM, and WT), providing an approach to estimate lake N removal through

remote sensing techniques[8,27,43]. Low-level occurrence of algae and the subsequent decomposition of the algae not only provides substrate N to stimulate denitrification, but also creates favorable conditions for sediment denitrification[44]. However, excessive algal decay can deplete DO and potentially disrupt the coupling of nitrification and denitrification and inhibit denitrification in sediments[43]. Meanwhile, warmer temperatures tend to linearly increase denitrification by enhancing microbial respiration rates, consuming oxygen and creating anaerobic conditions[8,45]. Therefore, a final additive conceptual model that includes a quadratic polynomial and linear terms links the dynamic N removal to Chl*a* and WT (Supplementary Fig. 1).

After building the conceptual remote sensing model, we used lake data from around the world to validate our model hypothesis. We conducted a meta-analysis on the relationship between Chl*a* and N concentration in 39,411 lakes worldwide in a database that was also used in the present study. The results showed a good regression relationship between the concentrations of Chl*a* and N in global lakes ($R^2 = 0.34$, $p < 0.01$), especially in shallow lakes, where a robust linear correlation between Chl*a* and N concentration existed[27,46]. Moreover, the concentrations of Chl*a* serve well as a proxy for N concentration even when lake algae growth is limited by the phosphorus (P)

concentration (where N:P > 22.4) ($R^2 = 0.25$, $n = 14{,}881$, $p < 0.01$, Supplementary Fig. 4). Then, we analyzed more than 20 previous studies which demonstrated that the rate of denitrification N removal from lakes can be well estimated from environmental factors that control the biogeochemical process of denitrification, especially water $NO_3^-$ and temperature (Supplementary Table 1). Therefore, our approach of using the remote sensing model that employs Chl*a* and temperature to estimate the rate of lake denitrification is robust.

We then parameterized and validated the conceptual model at the local and global scales. Lake Taihu was used to parametrize the remote sensing model as this lake covers nearly 90% of the range of variation in global lake N removal rates[27,47]. Next, the remote sensing model was validated at the global scale by comparing our global lake N removal estimations with the RivR-N model, which was empirically derived from water residence time and depth for estimating the lake N removal ratio across America and Europe[48]. In general, our estimates fit well with the results of the RivR-N model (see Supplementary Note 1).

**Estimation of N removal rate in global lakes.** Combining the developed remote sensing model and the datasets of Chl*a* and the annual surface water temperature of lakes, we estimated the N removal of global lakes in the HydroLAKES database, which includes 1.4 million lakes with a surface area >10 ha. Global lake Chl*a* concentration data were obtained from Filazzola et al.[49] which synthesized Chl*a* values of 11,959 freshwater lakes across 72 countries from 3322 published articles. If a lake had multiple observations, the average of the multiple observations was considered as the lake Chl*a* concentration in this study. The annual surface water temperature of global lakes was calculated based on the Copernicus Global Land Service database[50], which provides monthly global lake surface water temperature data at a 1-kilometer grid-scale. The monthly mean surface water temperatures of global lakes between September 2017 and August 2018 were averaged as the annual mean lake surface water temperature. As a lake's annual surface water temperature largely depends on the lake's latitude (Supplementary Fig. 5), we used the latitude data to estimate the annual surface water temperature of lakes in the Chl*a* dataset.

To constrain the scope of the application of our remote sensing model, we excluded deep lakes by comparing the thermocline depth of the lake (THER) with the average lake water depth. The THER was calculated based on the empirical relationship between the THER and lake surface area (SA) from 127 global lakes (Eq. (1))[51]. When the average depth of a lake is greater than the THER, we consider such lakes to be deep lakes, characterized by incomplete mixing and stratification of nutrients, temperature, and dissolved oxygen, and such a lake is not suitable to apply the remote sensing model[52]. Finally, the N removals of a total of 5768 lakes (Supplementary Fig. 6) were estimated, and these estimates are shown at the HydroSHEDS level-5 watershed scale.

$$\text{Log THER} = 0.185 \, \text{Log SA} + 0.842 \tag{1}$$

**The role of N removal in lake N budget.** The N removal ratio ($R_{ratio}$) is used to evaluate the role of N removal in lake N budget, defined as the proportion of lake N loading ($N_{loading}$) that can be permanently removed through $N_2$ emissions ($N_{output,removal}$).

$$R_{ratio} = \frac{N_{output, removal}}{N_{loading}} \tag{2}$$

The $N_{output,removal}$ was calculated using the permanent annual lake N removal rate ($N_{2,emission}$) and SA, in which the $N_{2,emission}$ was estimated using the remote sensing model we developed[27], and the lake SA was obtained from the HydroLAKES database.

$$N_{output, removal} = N_{2, emission} \times SA \tag{3}$$

The annual lake N loading ($N_{loading}$) was estimated as the sum of the lake N inflow through surrounding surface rivers and underground water loads ($N_{loading,inflow}$) and the direct atmospheric N deposition into the lake ($N_{loading,deposition}$):

$$N_{loading} = N_{loading, inflow} + N_{loading, deposition} \tag{4}$$

The N inflow from the surrounding surface rivers and underground water loads ($N_{loading,inflow}$) was considered as N entering a lake from upstream watershed soil N losses, which was estimated as the product of soil N input in a watershed and river loading coefficient ($a$). The river loading coefficient $a$ (0.3–0.5) represents the ratio of the soil N input in a watershed that can enter the lake through surface rivers and transport of underground water loads, which was based on literature values[2,8,15,53].

$$N_{loading, inflow} = N_{NANI} \times a \tag{5}$$

Here, watershed N input was considered the watershed Net Anthropogenic N Input ($N_{NANI}$), including atmospheric N deposition, input from N fertilizers, net import or export of N in agricultural commodities, and N fixation[54]. The global country-scale NANI for 2009 was obtained from Han et al.[55]. Considering that most lakes studied are distributed in the United States and China, we used a finer-resolution NANI database of the United States for 2012 and of China for 2007 for county-scale analysis[56,57].

Atmospheric N deposition ($N_{loading,deposition}$) data were obtained from Ackerman et al.[58], who used the GEOS-Chem Chemical Transport Model to estimate the wet and dry deposition of inorganic N globally at a spatial resolution of 2° × 2.5°. The product of the mean lake N deposition rate ($N_{deposition,mean}$) and lake SA was considered the total amount of N deposition directly from the atmosphere to the lake:

$$N_{loading, deposition} = N_{deposition, mean} \times SA \tag{6}$$

**Watershed N management and water quality improvement**
In this study, we developed a mass balance approach to link watershed N management and lake water quality improvement (Supplementary Fig. 1). Here, we considered the main N budget of the lake, including N loading, N outflow, N removal, and N storage. Theoretically, sediment N burial, resuspension, and diffusion also contribute to the lake N budget. However, they account for a small share of the lake N budget and are likely to balance each other in shallow lakes[59,60]. Therefore, we did not account for these N budget terms in the mass balance model. Following the equations below, we simulated how lake water quality improves with watershed N management, which includes: (i) how many years are required to achieve the water quality goal (≤1 mg N L$^{-1}$) at a fixed annual watershed N input reduction rate (2%, 5%, and 10% annual reduction compared to the N input in each preceding year); (ii) what is the reduction intensity of the watershed N input to achieve the water quality goal within limits of 10, 20, 30, and 40 years?

$$\frac{dN_{storage}}{dt} = \frac{dN_{loading}}{dt} - \frac{dN_{output, outflow}}{dt} - \frac{dN_{output, removal}}{dt} \tag{7}$$

$$\frac{dN_{loading}}{dt} = N_{loading} \times b \tag{8}$$

$$\frac{dN_{output, outflow}}{dt} = N_{ouput, outflow} \times b \tag{9}$$

$$\left(\frac{dN_{output, removal}}{dt}\right) / (N_{output, removal}) = \left(\frac{dN_{loading} + dN_{storage}}{dt}\right) / (N_{loading} + N_{storage}) \tag{10}$$

Where $N_{storage}$ is the lake N storage, calculated from the lake water volume multiplied by N concentration and can be obtained from the HydroLAKES database; $N_{loading}$ is the loading of N through rivers and underground water loads and the atmospheric N directly depositing into the lake surface, calculated according to Eq. (4); $N_{output,removal}$ is the lake N output that can be permanently removed through $N_2$ emission, estimated according to Eq. (3); $N_{output,outflow}$ is the lake N output through downstream outflow of rivers, and the current $N_{output,outflow}$ can be calculated according to mass balance (Eq. (11)); $b$ is the fraction of watershed N input that must be reduced to achieve the water quality goal. In our mass balance approach, the current lake N removal ($N_{output,removal}$) is related to the sum of the current lake N loading ($N_{loading}$) and storage ($N_{storage}$); thus, the dynamic N removal with watershed N reduction can be calculated according to the proportional relationship in Eq. (10).

$$N_{output, flow} = N_{loading} - N_{output, removal} \qquad (11)$$

In our simulation, as the targeted lake water quality ($c$) of $\leq 1$ mg N $L^{-1}$ was set for analysis[30], we could calculate the N storage in the target lake based on Eq. (12), where WV represents the lake water volume.

$$N_{storage, target} = WV \times c \qquad (12)$$

Finally, the magnitude of lake N loading that must be reduced to achieve the water quality goal within a given year can be calculated according to Eqs. (7)–(12). Specifically, in Supplementary Fig. 7, we used an example to illustrate how to simulate the effect of watershed N management on the lake N budget in the next year based on the lake N budget in each preceding year.

### Model validation and uncertainty analysis

We validated our mass balance approach for water quality improvement response to watershed N management in 49 HUC-8 watersheds of northeastern USA (Supplementary Fig. 8). The data of the 49 watersheds were obtained from the LAGOS-NE database[61], which synthesized 51,101 lakes Chl$a$ and N concentrations between 2001 and 2013. We selected the watersheds in the LAGOS-NE database where NANI decreased from 2002 to 2012 to verify the accuracy of the mass balance model, and finally obtained data from 49 HUC-8 watersheds. To reduce the uncertainty, we used the average of 2001, 2002, and 2003 as the value of 2002; and took the average of 2011, 2012, and 2013 as the value of 2012. Based on the TN and Chl$a$ concentrations in 2002, the N input reduction rate from 2002 to 2012, and the mass balance model, we estimated each watershed's TN concentrations in 2012. By comparing our estimated TN concentrations in 2012 with the field-measured TN concentrations in 2012, we evaluated the applicability of the mass balance model to simulate the response of lake water quality improvements to watershed N management. The coefficient of determination ($R^2$) between the model-predicted and measured TN concentrations reached 0.97, indicating a high predictive accuracy of the model (Supplementary Fig. 9).

Monte Carlo simulations were used to characterize the uncertainty associated with estimated lake N removal, lake N loading, and the required watershed N input reduction and times. A total of 1000 simulations for each input variable matchup of Chl$a$ and WT were conducted to estimate the uncertainties of the lake N removal for each lake. 1000 parameter sets for the lake N loading coefficient were sampled based on Monte Carlo to simulate the uncertainty of global lake N loading. 1000 simulations of lake N removal and N loading were sampled to identify the uncertainty of the required watershed N input reduction and times to achieve the water quality goal for each lake. Uncertainty ranges were provided with 95th confidence intervals from the Monte Carlo simulations.

## Data availability

The HydroLAKES dataset was retrieved from Global HydroLAB (https://wp.geog.mcgill.ca/hydrolab/hydrolakes/). The HydroSHEDS dataset was obtained from Global HydroLAB (https://www.hydrosheds.org/). The Chlorophyll and Water Chemistry databases were retrieved from Scientific Data (https://doi.org/10.1038/s41597-020-00648-2). The dataset of global watershed Net Anthropogenic Nitrogen Inputs (NANI) was synthesized from Geoderma (https://doi.org/10.1016/j.geoderma.2019.114066), Science of the Total Environment (https://doi.org/10.1016/j.scitotenv.2018.04.027), and Biogeochemistry (https://doi.org/10.1007/s10533-011-9606-y). The Lake Surface Water Temperature data was retrieved from the Copernicus Global Land Service (https://land.copernicus.eu/global/products/lswt). Source data are provided with this paper[62].

## Code availability

The Python (version 3.7) used for the present analysis is available from https://www.python.org/. The estimating of the lake N budget, potential strategies to improve lake water quality, and uncertainty analysis data used in this study are available on Figshare (https://doi.org/10.6084/m9.figshare.26509543)[62].

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

## Acknowledgements

This work was supported by the National Natural Science Foundation of China (42430706, 42177401 to Y.X.) and the National Key Research and Development Program of China (2021YFD1700802 to Y.X.).

## Author contributions

X.Y. (Xing Yan), Y.X., and X.Y. (Xiaoyuan Yan) conceived this study. X.Y. (Xing Yan) and Y.X. developed the water quality improvement models, analyzed the data, drew the graphs, and wrote the paper with direct contributions from X.Z., C.T., L.X., S.C., and X.Y. (Xiaoyuan Yan). All authors reviewed and commented on the revised manuscript.

## Competing interests

The authors declare no competing interests.
