## [Peer Review file · Nature Communications]

Coupling nitrogen removal and watershed management to improve global lake water quality

Corresponding Author: Professor Xiaoyuan Yan

Version 0:

Reviewer comments:

Reviewer #1

(Remarks to the Author)

The manuscript titled "Coupling Nitrogen Removal and Watershed Management to Improve Global Lake Water Quality" introduces work that couples nitrogen removal and watershed management to enhance the water quality of lakes worldwide, possessing significant academic value and reference significance. It is recommended that the authors make the following modifications before publishing in this journal.

Revision Suggestion:

1. Specifically state the number of lakes analyzed in the study (e.g., "This study analyzed data from a total of 6,870 lakes globally") and further refine the geographic distribution of these lakes. For instance, quantify the regional coverage by indicating the number of countries and their respective lake proportions in North America, Europe, and East Asia.
2. In the model validation section, in addition to qualitatively describing the fit between the model and existing data, provide specific quantitative metrics to assess model performance. For example, use the coefficient of determination (R^2) to quantify the degree of fit between predicted and measured values, clearly stating the R^2 value (e.g., "The coefficient of determination (R^2) between the model-predicted total nitrogen concentrations and measured values reached 0.85, indicating a high predictive accuracy of the model.").
3. When discussing water quality improvement strategies for different lake types (Type I, Type II, Type III), propose specific quantitative targets. For example, for Type II lakes, set a quantifiable goal such as "By reducing upstream nitrogen loads by XX%, it is projected that the total nitrogen concentrations in these lakes will decrease to ≤ 1 mg/L within the next YY years."

(Remarks on code availability)

Reviewer #2

(Remarks to the Author)

The manuscript presents some novel elements in examining the role of lakes in removing nitrogen (N) and determining timelines for achieving water concentrations of less than 1 mg/L. However, several key aspects remain unclear, particularly concerning the methodology. Authors have to work on better justification of many aspects of the manuscript, and they really need to support conclusions with results in more clear way. One of the crucial parts of the paper is the "remote sensing model," which lacks clarity and proper verification. The authors frequently refer to supplementary information and reference "29" without providing a clear explanation of the model itself or how it reproduces the presented results. There are several concerns regarding this model and its application:

(1) Model Scope and Upscaling: The model described in reference 29 was developed for a single lake but is applied here in a very arbitrary upscaling to 6,870 lakes. The lack of explanation or validation for this upscaling raises serious concerns about the model's robustness in this broader application. Furthermore, the model in reference 29 is designed specifically for "shallow" lakes, yet there is no clear definition of what constitutes "shallow" in the context of this paper. Additionally, most of the 6,870 lakes included in the study are not shallow, further emphasizing the need for clarification and justification.

(2) Nitrogen Prediction and Chlorophyll-a: The prediction of nitrogen concentrations in the lakes is based primarily on chlorophyll-a concentrations. However, chlorophyll-a is typically an indicator of phosphorus levels, not nitrogen, particularly

in phosphorus-limited systems like many of the 6,870 lakes studied. There is no justification provided for using a nitrogen-chlorophyll-a correlation in these phosphorus-limited systems. This approach is problematic unless the authors can provide strong justification for why this correlation holds in their case.

(3) Neglect of N Burial in Sediments: While the authors emphasize the importance of sediments in nitrogen processing, they fail to account for nitrogen burial in their overall budget. In any mass balance, the principle of in-out=0 applies, where "out" includes both outflow from the lake and burial in the sediments. This omission needs to be addressed to provide a complete picture of nitrogen cycling in the lakes.

(4) Oxygen Concentrations and N removal/production: Since the model strongly emphasizes the importance of oxygen, it would be crucial for the authors to establish or show from different dataset the relationship between oxygen levels in the lakes and nitrogen concentrations. Oxygen availability significantly influences processes like nitrification and denitrification, which are key to nitrogen removal. Additionally, the role of depth profiles in nitrogen production or removal should be explored further. Stratification and variations in oxygen concentrations with depth can significantly affect the nitrogen cycling within the lakes, and these dynamics should be considered, particularly if the authors intend to generalize their findings across multiple lakes.

Overall, while the paper explores an important topic with some novel insights, it requires substantial revision before it can be considered for publication. The authors must provide more detailed justifications for their approaches and ensure that the methodologies are clearly explained so that the results can be reproducible. I strongly recommend the authors revisit the remote sensing model, clarify its application, address the issues raised in the nitrogen-chlorophyll-a relationship and nitrogen burial.

Please note that if some tools, such as ChatGPT, are used for manuscript's writing, please make a note about that. Many sentences (e.g., lines 25-27; 66-69...) are without any useful information. They are very well structured, but do not contain any meaningful/important message.

(Remarks on code availability)

N/A

Reviewer #3

(Remarks to the Author)

The authors describe how they found differences in the efficiency of lakes to remove nitrogen (N) from the runoff waters and stored in the lake itself, by using a remote sensing model they developed. They suggest that this approach can efficiently help target measures to reduce external N loading to certain lakes, and save resources by not focusing on lakes that already meet the water quality criteria or are likely to reach this level with no management measures.

The topic of water management is very interesting and relevant, and it is important to constantly seek efficient approaches to improve the state of currently polluted lakes and prevent the degradation of those that are still in good state.

I find the approach of using a remote sensing model in this field fascinating and the authors have clearly done a thorough job in developing such a model. The manuscript is well written and concise. I was left with a few questions about the conclusions, though, and I think the authors could work on clarifying these.

It remains somewhat unclear what the authors think are the factors that determine the N removal efficiency of a given lake. They mention that lower N loading is linked to better N removal efficiency. But if this is the main factor, could it not be simply stated that lakes with low N loading do not require management measures in the catchment area, while lakes with high N loading do? In other words, what is the added value of the model to this knowledge? Are there perhaps lakes with high external N loading but also such a high N removal efficiency that, despite of the high loading, they do not require management in order to maintain good water quality? If so, what are the factors that determine this efficiency?

The authors may have discussed these questions briefly here and there, but the paper could be improved by giving more emphasis on these topics.

Please see the detailed comments in the attached file.

(Remarks on code availability)

Version 1:

Reviewer comments:

Reviewer #1

(Remarks to the Author)

(Remarks on code availability)

Reviewer #2

(Remarks to the Author)

The authors have successfully addressed the major comments raised in my initial review, and the revised manuscript has improved significantly as a result. The responses were clear and comprehensive, and the changes made to the text have enhanced both the clarity and scientific rigor of the study. I am satisfied with the revisions, and the manuscript is now in good shape. From my perspective, it is ready for acceptance in its current form.

(Remarks on code availability)

N/A

Reviewer #3

(Remarks to the Author)

I kindly thank the authors for their responses to the review comments and suggestions and implementing these into the manuscript. In my opinion, several key points are now presented more clearly and concisely and I think the manuscript improved.

What I would suggest the authors to still work on is the key message of this paper, which I still find somewhat ambiguous. The title or the paper, as well as the research questions stated in the introduction, suggest that the goals of this study were to show how the N removal efficiency of lakes is a key characteristic in defining their water quality – and thus their response to N loading reductions – and, by using the remote sensing model presented in this study, to identify the lakes in their dataset where N loading reductions lead to the most effective results. At the same time, the authors state: “Our study provides an effective tool to estimate watershed scale N removal in lakes through remote sensing techniques.”

Is the main purpose of this paper to offer a new tool (i.e., the remote sensing model the authors developed and used for this study) to be used for any area/watershed/lake in order to determine where N loading should be targeted to yield best results in the lake water quality improvement – hence the key message of this paper being the introduction of this method as a tool for lake management? Or is the main purpose to show, by using this remote sensing model, that there is variability in the N removal efficiency of lakes globally, and that this should be taken into account while planning N reduction strategies?

Both these aspects are relevant per se, but they are different approaches to the theme – one is to introduce a new tool/methodology (the remote sensing model for N budget in lakes), and the other is to present a natural phenomenon (variability in N removal efficiency between lakes) by using the tool in question. Is the scope and target of the paper clear enough? I do not mean to question the methodology or results of the paper, but I was left with the impression that the authors in fact aimed to answer more research questions than were presented in the introduction, and hence I feel like the common thread is sometimes lost along the manuscript.

However, this could also be just my different thinking or perhaps a conceptual misunderstanding, and if the Editor and other reviewers do not see any significant discontinuity in the manuscript, the authors can choose to ignore this feedback.

(Remarks on code availability)

Point-by-point Responses to Comments from the Reviewers

The authors would like to thank the efforts of the editorial office personnel and the reviewers for their critical comments and suggestions. We have tried our best to address those comments and have thoroughly revised our manuscript. We hope that our revision has properly addressed the concerns raised by the reviewers. As a result of the constructive comments from the reviewers, the quality of the manuscript has been substantially improved. For more details, please refer to the point-by-point responses to the comments listed below. Comments are numbered and in black text. Our responses are in blue, and line numbers refer to the lines in the document Revised_Manuscript_Tracked. All scientific changes are marked with a yellow color background in the revised manuscript, and quoted texts from the revised manuscript are in purple.

Responses to Reviewer #1's Comments

[General comments] *The manuscript titled "Coupling Nitrogen Removal and Watershed Management to Improve Global Lake Water Quality" introduces work that couples nitrogen removal and watershed management to enhance the water quality of lakes worldwide, possessing significant academic value and reference significance. It is recommended that the authors make the following modifications before publishing in this journal.*

[Response] Thank you for your positive feedback on our manuscript. We greatly appreciate your suggestions. We have carefully addressed each comment and made thorough revisions.

Detailed comments

[Comment 1] *Specifically state the number of lakes analyzed in the study (e.g., "This study analyzed data from a total of 6,870 lakes globally") and further refine the geographic distribution of these lakes. For instance, quantify the regional coverage by indicating the number of countries and their respective lake proportions in North America, Europe, and East Asia.*

[Response] Thank you for your valuable suggestion. In the revised manuscript, we have stated and refined the geographic distribution as follows:

“These lakes mainly include 5,515 lakes in North America, 187 lakes in Europe, and 59 lakes in Asia.” (lines 711-712)

In addition, we have added new analyses to Supplementary information to quantify the eutrophic states of the lakes in this study based on the literature¹ as follows:

Supplementary Table 1. Distribution of lake trophic state in this study.

	Oligotrophic Chla < 2 $\mu\text{g L}^{-1}$	Mesotrophic 2 \leq Chla < 7 $\mu\text{g L}^{-1}$	Eutrophic 7 \leq Chla < 30 $\mu\text{g L}^{-1}$	Hypereutrophic Chla \geq 30 $\mu\text{g L}^{-1}$
Lake number count	474	2,701	1,862	731
Trophic state distribution (%)	8.22	46.83	32.28	12.67

[Comment 2] *In the model validation section, in addition to qualitatively describing the fit between the model and existing data, provide specific quantitative metrics to assess model performance. For example, use the coefficient of determination (R^2) to quantify the degree of fit between predicted and measured values, clearly stating the R^2 value (e.g., "The coefficient of determination (R^2) between the model-predicted total nitrogen concentrations and measured values reached 0.85, indicating a high predictive accuracy of the model.")*

[Response] Thank you for your valuable suggestion. In the model validation section, we have added specific quantitative metrics to assess model performance as follows:

“The coefficient of determination (R^2) between the model-predicted and measured TN concentrations reached 0.97, indicating a high predictive accuracy of the model (Extended Data Fig. 4).” (lines 573-576)

[Comment 3] *When discussing water quality improvement strategies for different lake types (Type I, Type II, Type III), propose specific quantitative targets. For example, for Type II lakes, set a quantifiable goal such as "By reducing upstream nitrogen loads by XX%, it is projected that the total nitrogen concentrations in these lakes will decrease to ≤ 1 mg/L within the next YY years."*

[Response] Thanks. We have proposed specific quantitative targets when discussing water quality improvement strategies as follows:

“Out of those lakes, 5,036 lakes have already met the water quality goal of ≤ 1 mg N L⁻¹ (with a mean TN concentration of 0.55 mg N L⁻¹), and there is no need to reduce watershed N input beyond the current N input level within these lakes.” (lines 251-254)

“The other 198 type I lakes do not yet meet the water quality goal, but the capacity for N removal is greater than the N loading rate (with a mean N removal ratio of 218%), and there is also no need to reduce watershed N input for these lakes.” (lines 254-257)

“Type II lakes (484 lakes) could achieve the water quality goal within 20 years by reducing an average of 13% lake N loading received from upstream in the watershed.” (lines 258-260)

Responses to Reviewer #2's Comments

[General comments] *The manuscript presents some novel elements in examining the role of lakes in removing nitrogen (N) and determining timelines for achieving water concentrations of less than 1 mg/L. However, several key aspects remain unclear, particularly concerning the methodology. Authors have to work on better justification of many aspects of the manuscript, and they really need to support conclusions with results in more clear way. One of the crucial parts of the paper is the "remote sensing model," which lacks clarity and proper verification. The authors frequently refer to supplementary information and reference "29" without providing a clear explanation of the model itself or how it reproduces the presented results.*

[Response] We would like to express our gratitude for your comprehensive review and constructive feedback. We appreciate your recognition of the novel elements in our work regarding the role of lakes in nitrogen (N) removal and our efforts to define timelines for achieving N concentrations of less than 1 mg/L.

We absolutely agree with you that it is very important to explain and validate the rationale for using remote sensing models to estimate lake N removal rate through denitrification. As many studies have shown, lake N removal through bacterial denitrification is the primary pathway for permanent reactive N removal from aquatic ecosystems by converting nitrate (NO_3^-) to gaseous nitrogen (N_2)². Therefore, quantifying lake N removal through denitrification is critical for watershed N management to improve lake water quality. However, accurately quantifying the magnitude of N removal through denitrification has long been challenged due to the high background concentrations of atmospheric N_2 , especially due to the difficulty in obtaining parameters and data at large regional scales^{3,4}.

Based on our laboratory's previous work over the past years and extensive literature analysis⁵⁻⁷, we found that watershed-scale estimations of N removal in lakes may be achieved by linking the biogeochemical variables that control N removal (e.g., concentrations of dissolved carbon, N, and oxygen (DO), and temperature of lake water (WT)) with remote sensing data, e.g., chlorophyll-*a* (Chl*a*), chromophoric dissolved organic matter (CDOM), and WT. Following this scientific hypothesis, we carefully constructed a remote sensing model for estimating regional scale lake N removal from the aspects of experimental design, data collection, model construction, and model verification. We published the remote sensing model as a research article in the journal *Environmental Research Letters* (DOI 10.1088/1748-9326/ad1f05); the remote sensing model provides a methodological tool for estimating lake N removal on a global scale.

However, our focus in this article is on strategies for watershed N management to improve lake water quality and achieve the UN's SDG. To maintain focus and brevity, we only briefly described the methodology of the remote sensing model in the previous version of this manuscript, resulting in a lack of clarity and verification. In the revised manuscript, we have clarified the methodology and provided a more thorough explanation of how the model reproduces the results, as well as minimized reliance on supplementary information and reference "29", ensuring that all essential details are directly integrated into the main text as follows (lines 416-453):

“Overview of the remote sensing model for lake N removal estimation

The remote sensing model used for estimating lake N removal is described in detail in our earlier study⁸. In principle, most of the environmental variables that control lake N removal (e.g., dissolved carbon and N, DO, and WT) can be directly or indirectly derived from remote sensing data (e.g., Chl_a, CDOM, and WT), providing an approach to estimate lake N removal through remote sensing techniques^{5,7,8}. First, low-level occurrence of algae and the subsequent decomposition of the algae not only provides substrate N to stimulate denitrification, but also create favorable conditions for sediment denitrification⁹. However, excessive algal decay can deplete DO and potentially disrupt the coupling of nitrification and denitrification and inhibit denitrification in sediments⁷. Second, warmer temperatures tend to linearly increase denitrification by enhancing microbial respiration rates, consuming oxygen and creating anaerobic conditions^{5,10}. Therefore, a final additive conceptual model that includes a quadratic polynomial and linear terms links the dynamic N removal to Chl_a and WT (Extended Data Fig. 1).

After building the conceptual remote sensing model, we used lake data from around the world to validate our model hypothesis. First, we conducted a meta-analysis on the relationship between Chl_a and N concentration in 39,411 lakes worldwide in a database that was also used in the present study. The results showed a good regression relationship between the concentrations of Chl_a and N in global lakes ($R^2 = 0.34$, $p < 0.01$), especially in shallow lakes, where a robust linear correlation between Chl_a and N concentration existed^{8,11}. Moreover, the concentrations of Chl_a serve well as a proxy for N concentration even when lake algae growth is limited by the phosphorus (P) concentration (where N:P > 22.4) ($R^2 = 0.25$, $n = 14,881$, $p < 0.01$, Supplementary Fig. 2). Then, we analyzed more than 20 previous studies which demonstrated that the rate of denitrification N removal from lakes can be well estimated from environmental factors that control the biogeochemical process of denitrification, especially water NO₃⁻ and temperature (Table S2). Therefore, our approach of using the remote sensing model that employs Chl_a and temperature to estimate the rate of lake denitrification is robust.

We then parameterized and validated the conceptual model at the local and global scale. Lake Taihu was used to parametrize the remote sensing model as this lake covers nearly 90% of the range of variation in global lake N removal rates^{8,12}. Next, the remote sensing model was validated at the global scale by comparing our global lake N removal estimations with the RivR-N model, which was empirically derived from water residence time and depth for estimating the lake N removal ratio across America and Europe¹³. In general, our estimates fit well with the results of the RivR-N model (see Supplementary Note 1 and our previous article⁸)”

In addition to clearly expressing how the model reproduces the results presented, we also tried our best to address your other concerns regarding the application of the remote sensing model in deep lakes, the Chl_a-N relationship in P-limited lakes, and N burial. Briefly, we have reanalyzed the data by excluding deep lakes, strictly limiting the scope of application in shallow lakes of our remote sensing model. The results from

the new data still support our previous conclusions. Moreover, we have added a new analysis to show the relationship between the concentrations of Chla and N and DO using the same dataset, demonstrating that Chla can still serve as the proxy for N in P-limited lakes. Lastly, we have conducted literature analysis to discuss how N burial affects the robustness of our model outcomes.

We believe that we have addressed your concerns in this revision, and the manuscript is much stronger because of your suggestions. For more detailed revisions and responses, please see below.

Detailed comments

[Comment 1] *Model Scope and Upscaling: The model described in reference 29 was developed for a single lake but is applied here in a very arbitrary upscaling to 6,870 lakes. The lack of explanation or validation for this upscaling raises serious concerns about the model's robustness in this broader application. Furthermore, the model in reference 29 is designed specifically for "shallow" lakes, yet there is no clear definition of what constitutes "shallow" in the context of this paper. Additionally, most of the 6,870 lakes included in the study are not shallow, further emphasizing the need for clarification and justification.*

[Response] We acknowledge the limitations of our model in terms of scope and upscaling. As stated above, the rationale for using a single lake for broad upscaling estimates of N removal rates in lakes was carefully verified, as well as verified in our article published in the journal *Environmental Research Letters* (DOI 10.1088/1748-9326/ad1f05). Briefly, the data for remote sensing model parameterization in Lake Taihu, a typical lake, can cover nearly 90% of the range of variations of the global lake N removal rates (Response Fig. 1). Thus, we directly adopt the model parameter in Lake Taihu for the preliminary validation.

Response Fig. 1 The variation of global lake N removal rates and the range of the measured N removal rates of Lake Taihu to develop the remote sensing model. The measured data in Lake Taihu can cover nearly 90% of the range of variations of global lake N removal rates. The global lake N removal data were synthesized from Qin et al.¹².

Meanwhile, in the revised version of the manuscript, we have removed deep lakes for analysis to severely limit the scope of our model by comparing the lake average depth and thermocline depth¹⁴. According to the data availability of lake attribute in our database, the thermocline depth of lake (THER) was calculated based on the empirical relationship between the THER and lake surface area (SA) from 127 global lakes¹⁵:

$$\text{Log THER} = 0.185 \text{ Log SA} + 0.842$$

When the average depth of a lake is greater than the THER, we consider such lakes to be deep lakes, characterized by incomplete mixing and stratification of nutrients, temperature, and dissolved oxygen. Therefore, we have deleted deep lakes in the revised manuscript as such lakes may not be suitable for remote sensing to estimate lake denitrification rates regarding the differences in lake properties between surface and deep water. The results from the new data still support our conclusions, and all figures and data have been revised throughout the revised manuscript. In fact, the average depth of our latest data for 5,768 lakes is only 4.2 m, which is consistent with the scope of shallow lakes. We have clearly explained how to restrict our remote sensing model to shallow lakes in the revised manuscript as follows (lines 471-481):

“To constrain the scope of the application of our remote sensing model, we excluded deep lakes by comparing the thermocline depth of the lake (THER) with the average lake water depth. The THER was calculated based on the empirical relationship between the THER and lake surface area (SA) from 127 global lakes (equation (1))¹⁵. When the average depth of a lake is greater than the THER, we consider such lakes to be deep lakes, characterized by incomplete mixing and stratification of nutrients, temperature, and dissolved oxygen, and such a lake is not suitable to apply the remote sensing model¹⁴. Finally, the N removals of a total of 5,768 lakes (Extended Data Fig. 3) were estimated, and these estimates are shown at the HydroSHEDS level-5 watershed scale.

$$\text{Log THER} = 0.185 \text{ Log SA} + 0.842 \quad (1)”$$

Lastly, our estimations of global lake N removals using the remote sensing model can be validated by the RivR-N model using the new version of data without deepwater lakes. In general, our estimates fit well with the results of the RivR-N model (Supplementary Fig. 6 and Note 1). Therefore, after strict constraints on the application of our remote sensing model in shallow lakes and careful validation with other model, we believe that the estimates of lake N removal produced by our remote sensing model are robust.

[Comment 2] *Nitrogen Prediction and Chlorophyll-a: The prediction of nitrogen*

concentrations in the lakes is based primarily on chlorophyll-a concentrations. However, chlorophyll-a is typically an indicator of phosphorus levels, not nitrogen, particularly in phosphorus-limited systems like many of the 6,870 lakes studied. There is no justification provided for using a nitrogen-chlorophyll-a correlation in these phosphorus-limited systems. This approach is problematic unless the authors can provide strong justification for why this correlation holds in their case.

[Response] We agree that P is a limiting factor for algal growth in lakes. However, many studies have also shown that N is also a limiting factor for algal growth in lakes, especially in shallow eutrophic lakes^{11,16}. To further enhance the robustness of our remote sensing model, we first conducted a meta-analysis on the relationship between Chla and N concentration in 39,411 lakes worldwide without distinguishing N-limitation or P-limitation using the same dataset with this present study. The regression results show a good relationship between the concentrations of Chla and TN ($R^2 = 0.34$, $p < 0.01$, Response Fig. 2).

Response Fig. 2 The relationship between Chla and TN concentrations for 39,411 lakes worldwide without distinguishing N-limitation or P-limitation. Data were synthesized from the same dataset as those in this present study¹⁷.

To further discuss the role of N on Chla in P-limited systems, we have added a new analysis to discuss the relationship between Chla and TN only in P-limited lakes with N:P ratio >22.4 . The results suggest that Chla can still be used as a proxy for N concentration even in P-limited lakes ($R^2 = 0.28$, $p < 0.01$, $n = 14,881$, Supplementary Fig. 2). Moreover, previous studies have reported that shallow lakes are more

susceptible to eutrophication and both N and P are important for algal growth¹⁶. Therefore, we believe that our remote sensing model assumption that Chl a can be used as the proxy for TN is reasonable after carefully re-examining the dataset and restricting our model application to shallow lakes only. The relationship between the concentrations of Chl a and N in P-limited lakes has been added as Supplementary Fig. 2 in the revised manuscript.

Supplementary Fig. 2 The relationship between Chl a and TN concentrations for 14,881 P-limited lakes with N:P ratio >22.4 . Data were synthesized from the same dataset as those in this present study¹⁷.

[Comment 3] *Neglect of N Burial in Sediments: While the authors emphasize the importance of sediments in nitrogen processing, they fail to account for nitrogen burial in their overall budget. In any mass balance, the principle of in-out=0 applies, where "out" includes both outflow from the lake and burial in the sediments. This omission needs to be addressed to provide a complete picture of nitrogen cycling in the lakes.*

[Response] We agree that input-output should equal zero in any mass balance. In the complete picture of N cycling in the lakes, many studies in shallow lakes have shown that the amount of N Burial in sediments is roughly balanced by the amount of N return to overlying water through sediment resuspension and dissolved N diffusion fluxes across the sediment-water interface¹⁸⁻²⁰.

Sediment Resuspension: We have added a literature analysis to highlight the importance of sediment resuspension in lakes. The results showed that almost 80-90% of bottom deposits in lakes would be resuspension under the action of peripheral wind/wave attack, random redistribution of sediment, and complete mixing (Response Table 1). In one of the core lakes studied in this present study (Lake Taihu in China), a

previous study suggested that sediments resuspended and buried in almost equal amounts²¹ (Response Fig. 3). Even in deep lakes, bottom deposits in deeper areas may transport shallow areas by turbidity currents, nepheloid layers or even density currents, and hence resuspension back into the overlying water¹⁸.

Response Table 1. Ratios of resuspension from several lakes worldwide.

Lake	Location	Resuspension flux ($\text{g m}^{-2} \text{d}^{-1}$)	gross flux (%)	Comments	Reference
Blue Chalk	Canadian	0.85	90	Annual average of 4 years; average of 10 stations	Dillon et al. ²²
Erie	USA	7-112	88-99	Fall overturn; stations at 9, 25, 40 m	Bloesch ²³
Lough Neagh	Ireland	21	90	Annual average; single site at 14 m	Flower ²⁴
Ontario	Canadian	15	85	Annual average; 23 m site	Rosa ²⁵
Päijänne	Finland	0.6-1.4	51-79	Annual average; range of 4 stations, 18-49 m	Kansanen et al. ²⁶
Wingra	USA	11	82	Average of 11 sample dates; duplicate traps, center of lake	Gasith ²⁷
36 Florida lakes	USA	/	80	Average of 36 lakes	Bachmann et al. ²⁸
Fish ponds	Israel	/	60-90	Average of 9 ponds with 2.0-4.5 ha	Avnimelech et al. ²⁹
Erken	Sweden	/	84-90	9 m	Weyhenmeyer et al. ³⁰
Limmaren	Sweden	/	83-94	4.6 m	Weyhenmeyer et al. ³⁰

Response Fig. 3 Annual net N flux after sediment resuspension minus burial in Lake Taihu.

Diffusion fluxes: Meanwhile, regeneration and transfer processes of N across the sediment-water interface are crucial to N release from sediments to the water column, more specifically for the NH_4^+ -N release³¹. Theoretically, the diffusion process of pore water N is based on concentration gradients between pore water and overlying water, and the regeneration process from sediments can be released to the water column²⁰. Studies have assumed that the balance between sedimentation and burial rates is equivalent to the fluxes of dissolved constituents¹⁹. Moreover, this internal N release from sediment will increase during the process of lake water quality improvement as the gradient between the N concentration in the sediment pore water and the overlying water becomes larger.^{32,33} Overall, the N released to the water column through diffusion from sediment can offset part of the N flux buried by sediment.

Model Uncertainty Analysis and Validation: We have conducted an uncertainty analysis to discuss the impact of these items of a small share in lake N budget that are not included in this present study (e.g., burial, resuspension, diffusion, bioturbation) on the estimated results of water quality improvement. The results show that these smaller N budgets have little impact on the model results compared to denitrification and N outflow. Moreover, we validated the mass balance model in 49 HUC-8 watersheds of the northeastern USA based on the field-measured data of the 10-year step length. The coefficient of determination (R^2) between the model-predicted and measured TN concentrations reached 0.97, indicating a high predictive accuracy of the model (Extended Data Fig. 4). Therefore, we believe that the results of our model regarding water quality improvement remain robust overall.

In summary, we did not count N burial, resuspension, and diffusion in the lake N budget, which are smaller share items and are likely to balance each other, as stated in the revised manuscript as follows:

“Theoretically, sediment N burial, resuspension, and diffusion also contribute to the lake N budget. However, they account for a small share of the lake N budget and

are likely to balance each other in shallow lakes. Therefore, we did not account for these N budget terms in the mass balance model.” (lines 521-524)

[Comment 4] *Oxygen Concentrations and N removal/production: Since the model strongly emphasizes the importance of oxygen, it would be crucial for the authors to establish or show from different dataset the relationship between oxygen levels in the lakes and nitrogen concentrations. Oxygen availability significantly influences processes like nitrification and denitrification, which are key to nitrogen removal. Additionally, the role of depth profiles in nitrogen production or removal should be explored further. Stratification and variations in oxygen concentrations with depth can significantly affect the nitrogen cycling within the lakes, and these dynamics should be considered, particularly if the authors intend to generalize their findings across multiple lakes.*

[Response] We agree that dissolved oxygen concentrations are crucial to our remote sensing models because they influence processes like nitrification and denitrification. However, dissolved oxygen affects nitrification and denitrification processes more by directly affecting enzyme concentrations and microbial activities (e.g. anaerobic environments tend to favor anaerobic denitrifying bacteria to carry out denitrification processes), rather than mediating N removal/production processes by affecting N concentrations (lines 422-427). Therefore, one of the assumptions of our model is to estimate the denitrification rate by linking Chl a and dissolved oxygen concentrations. Moreover, the relationship between dissolved oxygen and N concentrations in lakes can be reflected by the relationship between dissolved oxygen and Chl a concentrations, because dissolved oxygen and Chl a in lakes are widely reported to be collinear from tropical lakes to subtropical lakes and temperate-zone lakes³⁴⁻³⁶. Meanwhile, we found that even without DO, the denitrification rate of water bodies can be well estimated by relying on N concentrations and temperature (Supplementary Table 1).

Supplementary Table 1. Aquatic ecosystem denitrification N removal estimated by environmental factors.

Variables	R^2/r	n	Types	Location	Reference
Water NO ₃ ⁻ and, water temperature	$R^2 = 0.86, P = 0.000$	18	River, pond, and reservoir	Jurong reservoir watershed, eastern China	Li et al. ⁶
Water NO ₃ ⁻ and water temperature	$R^2 = 0.85, P = 0.000$	36	River	Lake Taihu basin	Zhao et al. ⁵
Water NO ₃ ⁻ and water temperature	$R^2 = 0.78, P = 0.000$	21	River	Lake Taihu basin	Zhao et al. ⁵
Water NO ₃ ⁻	$R^2 = 0.86, P = 0.000$	136	Oceans, estuaries, lakes, and rivers	A global meta-analysis	Piña-Ochoa and Álvarez-Cobelas ³⁷
Water NO ₃ ⁻	$R^2 = 0.53$	23	Lake	Lake Shelbyville, Mississippi River basin	David et al. ³⁸
Water Chl a	$R^2 = 0.52, P = 0.000$	30	Lake	New Zealand	Bruesewitz et al. ³⁹
Water TN	$R^2 = 0.51, P = 0.000$	30	Lake	New Zealand	Bruesewitz et al. ³⁹
Water NO ₃ ⁻	$r = 0.51, P < 0.05$	15	Lake	Lake Bosten (China)	Jiang et al. ⁴⁰
Water NO ₃ ⁻	$R^2 = 0.87, P < 0.05$	21	Lake	Swiss	Müller et al. ⁴¹
Water NO ₃ ⁻	$r = 0.66, P < 0.05$	10	Lake	Yangtze River basin	Liu et al. ⁴²
Water NO ₃ ⁻ and water temperature	$R^2 = 0.48, P < 0.05$	11	Lake	Pyrenees	Palacin-Lizarbe et al. ⁴³
Water NO ₃ ⁻	$r = 0.53, P < 0.01$	24	Lake	Meiliang Bay and Inner Bay, Lake Taihu basin	Zhong et al. ⁴⁴
Water DIN and water temperature	$R^2 = 0.58, P < 0.01$	20	Drainage ditches	Zhushanwan watershed, Lake Taihu basin	She et al. ⁴⁵
Water NO ₃ ⁻	$r = 0.61, P < 0.01$	48	Lake	Lake Taihu	Liu et al. ⁴²
Water NO ₃ ⁻	$r = 0.78, P < 0.01$	75	Lake	Poyang Lake	Yao et al. ⁴⁶
Water NO ₃ ⁻	$r = 0.79, P < 0.01$	90	Lake	Poyang Lake	Zhang et al. ⁴⁷
Water DIN	$R^2 = 0.76$	13	River	Jiulong River, southeast China	Chen et al. ⁴⁸

Additionally, by comparing the average lake depth to the thermocline depth, we have removed deep lakes that are likely stratified and have variations in temperature with depth. Lake thermal stratification controls water mixing, nutrient cycles, and the vertical distribution of dissolved oxygen^{14,49}. Therefore, dissolved oxygen stratification is almost non-existent in the shallow lakes analyzed in this study, refining the remote sensing model to a suitable application case.

[Comment 5] *Overall, while the paper explores an important topic with some novel insights, it requires substantial revision before it can be considered for publication. The authors must provide more detailed justifications for their approaches and ensure that the methodologies are clearly explained so that the results can be reproducible. I strongly recommend the authors revisit the remote sensing model, clarify its application, address the issues raised in the nitrogen-chlorophyll-a relationship and nitrogen burial.*

[Response] We are grateful for your insightful feedback. To address the concerns raised about the remote sensing model and its application in nitrogen-chlorophyll-a relationship and nitrogen burial. We have revisited the relationship between Chl_a and N in remote sensing models, especially in P-limited lakes. We have excluded data from deep lakes and discussed how burial and dissolved oxygen stratification affect the robustness of our model outcomes. Overall, we believe that we have adequately addressed these concerns, and the revised manuscript become stronger now as a result of your valuable suggestions.

[Comment 6] *Please note that if some tools, such as ChatGPT, are used for manuscript's writing, please make a note about that. Many sentences (e.g., lines 25-27; 66-69...) are without any useful information. They are very well structured, but do not contain any meaningful/important message.*

[Response] Thank you for your valuable feedback. We would like to clarify that no AI tools, including ChatGPT, were used in the writing of this manuscript. Some sentences were intended to provide context or transitions that were delicately crafted. However, we understand that they may appear less informative, and we have revised these sections to ensure they convey clear and meaningful information as follows:

Original text: “We found that watershed N input and lake water quality are nonlinearly related and influenced by variability in lake N removal efficiency.”

Revised as: “We found that watershed N input reduction and lake water quality improvement are nonlinearly related and influenced by the efficiency of lake N removal.” (lines 27-29)

Original text: “In addition, it is a challenge to accurately quantify the varied lake N removal rates due to heterogeneous environmental factors at the landscape scale, impeding the development of a lake-specific water quality improvement strategy.”

Revised as: “In addition, accurately quantifying the varied N removal rates in lakes remains challenging due to heterogeneous environmental factors at the landscape scale^{2,50,51}, impeding the development of targeted water quality improvement strategies for individual lakes.” (lines 70-74)

Original text: “However, the heterogeneity in the distribution of global lake N removal is different from the heterogeneity of lake N loading and N concentration in lake waters due to nonlinear relationships between these extensively interacting N budget items.”

Revised as: “However, the global distribution of lake N removal exhibits a distinct pattern of heterogeneity compared to the N loading and N concentration in lake waters, driven by the nonlinear relationships among these extensively interacting items of the N budget.” (lines 101-105)

Original text: “Therefore, it is possible that lake N removal is greater than lake N loading as some of the N removed comes from the N stored in the lake.”

Revised as: “This suggests that lake N removal can exceed N loading, as part of the removed N may originate from N already stored in the lake.” (lines 135-136)

Original text: “Given that lake N concentration and water quality improvement efficiency vary among lakes, applying the same watershed N management strategy focused on N input reduction across global lakes leads to variable results in water quality improvement.”

Revised as: “Given that lake N concentration and water quality improvement efficiency vary among lakes^{4,52}, applying uniform N input reduction strategies yields differing outcomes globally^{53,54}.” (lines 226-229)

Additionally, we have tried our best to revise the grammar to make our manuscript more concise and readable (tracked but without a yellow color background throughout the revised manuscript). We hope that the revised manuscript will meet the high language standards of the journal.

Responses to Reviewer #3’s Comments

[General comments] *The authors describe how they found differences in the efficiency of lakes to remove nitrogen (N) from the runoff waters and stored in the lake itself, by using a remote sensing model they developed. They suggest that this approach can efficiently help target measures to reduce external N loading to certain lakes, and save resources by not focusing on lakes that already meet the water quality criteria or are likely to reach this level with no management measures.*

The topic of water management is very interesting and relevant, and it is important to constantly seek efficient approaches to improve the state of currently polluted lakes and prevent the degradation of those that are still in good state.

I find the approach of using a remote sensing model in this field fascinating and the authors have clearly done a thorough job in developing such a model. The manuscript is well written and concise.

I was left with a few questions about the conclusions, though, and I think the authors could work on clarifying these.

It remains somewhat unclear what the authors think are the factors that determine the N removal efficiency of a given lake. They mention that lower N loading is linked to better N removal efficiency. But if this is the main factor, could it not be simply stated that lakes with low N loading do not require management measures in the catchment area, while lakes with high N loading do? In other words, what is the added value of the model to this knowledge? Are there perhaps lakes with high external N loading but also such a high N removal efficiency that, despite of the high loading, they do not require management in order to maintain good water quality? If so, what are the factors that determine this efficiency?

The authors may have discussed these questions briefly here and there, but the paper could be improved by giving more emphasis on these topics.

[Response] Thank you for your thoughtful and encouraging comments. We greatly appreciate your recognition of the relevance and novelty of our approach, as well as your acknowledgment of the thoroughness of our model development. Your positive feedback motivates us to continue refining our work.

We agree with you that “lakes with lower N loading are linked to better N removal efficiency, and low N loading do not require management measures in the catchment area, while lakes with high N loading do”. This is indeed basic. However, our model allows us, for the first time, to quantify the relationship between the amount of the reduction in watershed N input and improved lake quality rather than just qualitatively indicating which lakes need management. Meanwhile, we focus more on the lake that needs management and analyze how much, when, and where to prioritize watershed N input reductions in the basin to improve water quality and achieve the SDG targets. Finally, as we state below, both the external lake N loading and the biochemistry and environmental factors affect the activity of the denitrifying microbial process. Some studies have shown that N removals are more efficient in warmer, shallower, smaller, and more evenly mixed lakes^{5,51,55}. Due to the limitations of the data on lake properties and the focus of the manuscript, such cases are not discussed carefully. However, your insights have given us great inspiration, and we will continue to explore this issue in depth in future work.

To clearly show the added value of the model to the knowledge of lake N budget and watershed N management to improve water quality goal. We have added an expanded discussion about the implications of our findings as follows:

“Our study provides an effective tool to estimate watershed scale N removal in lakes through remote sensing techniques. This allows us, for the first time, to quantify the relationship between the amount of the reduction in watershed N input and improved lake quality on a global scale. The results suggest the need for target measures to reduce external N loading to certain lakes, and save resources by not focusing on lakes that already meet the water quality criteria or are likely to reach this level with no management measures. Moreover, our study provides a specific strategy for watershed management of 5,768 global lakes to achieve the UN’s SDG goals for water quality improvement.” (lines 289-300)

Detailed comments

Ln 22. Enhancing the water quality of what? Waters that flow through the lakes and finally enter the sea? Phrased this way, the sentence sounds iterative, as if lakes played an important role in improving their own water quality. It is important to clarify what system(s) the lakes are thought to serve by removing nitrogen loading from the watersheds.

[Response] We agree with the comment you pointed out. It refers to enhancing the water quality that is temporarily stored in the lake and finally flows out of the lake and enters the sea. In fact, within a hydraulic residence time, the water stored in the lake will be completely replaced/flushed by water from the upstream river. During this cycle, the lake removes N, which is temporarily stored in the lake through denitrification and eventually flows out of the lake into the sea. We have revised the manuscript as follows:

“Lakes play a crucial role in removing nitrogen (N) loads from contributing watersheds and enhancing the quality of water that eventually flows out into the sea.” (lines 22-24)

Ln 27-29. I find this sentence somewhat confusing as well. Is the N removal efficiency characteristic to the lake basin or the surrounding watershed? In the second sentence of the abstract, it is stated that the efficacy of N removal varies among lakes, but here it sounds like it's the watersheds that are thought to affect this efficacy. Where is this improvement thought to take place? In the lake or the waters downstream (ultimately the sea)?

[Response] Thank you for your valuable comment. As stated above, N removal occurs in lakes, which improves the water in the lake through denitrification. However, the water in the lake is only temporarily stored in the lake and will finally flow out of the lake into the sea in a hydraulic residence period.

According to the first-order kinetic equation, the N removal efficiency of a lake

depends on both the N concentration in the lake water (i.e., the amount of substrate available for reaction) and the biochemistry and environmental factors that affect the activity of the denitrifying microbial process (e.g., water temperature). However, the N content in the lake mainly depends on the amount of N input from the basin's upper reaches. Therefore, it can be said that the amount of N input in the watershed and the characteristics of the lake determine the differences in lake N removal efficiency. To avoid this confusion, we have revised the manuscript as follows:

“Nevertheless, the N removal efficacy varies widely among lakes due to different watershed N input rates and lake characteristics⁵⁶⁻⁵⁸, complicating decision-making on watershed N management.” (lines 24-26)

Ln 90-94. This sentence is somewhat confusing - I suggest to rephrase it for clarity.

[Response] We have split this sentence into two sentences and rephrased it as follows:

“The anticipated hotspots of lake N loading, N concentration, and N removal rate simultaneously exist in areas with high anthropogenic reactive N input (Fig. 1a). Higher N input to watershed soil facilitates N transport into the lake through runoff, leading to elevated lake water N concentrations and increased N removal capacity.” (lines 95-101)

Ln 117. remove this word

[Response] Thanks, removed accordingly. (line 128)

Ln 128-131. Isn't this basically the same as "lakes with low external N loading have better water quality", which is quite elementary?

[Response] We agree that most of these lakes with an N removal ratio of >100% are lakes with low external N loading and better water quality. We re-examined our data and found that among the 2,049 lakes with N removal ratio >100%, 1,835 lakes have TN concentrations $\leq 1 \text{ mg N L}^{-1}$, with a mean of 0.52 mg N L^{-1} . However, the remaining 214 lakes have N removal ratios >100%, but their water quality was still seriously polluted, with TN concentrations $> 1 \text{ mg N L}^{-1}$ and a mean of 3.75 mg N L^{-1} .

The difference between these two types lakes is the watershed N input (NANI) in basins with lake N removal ratios >100% and TN concentrations $\leq 1 \text{ mg N L}^{-1}$ (mean of $2,906 \text{ kg NANI km}^{-2} \text{ yr}^{-1}$) are lower than those in basin with lake N removal ratios >100% and TN concentrations $> 1 \text{ mg N L}^{-1}$ (mean of $3,639 \text{ kg NANI km}^{-2} \text{ yr}^{-1}$) (Response Fig. 4). Moreover, we found that the NANI in basin with lake N removal ratio >100% and TN concentrations $> 1 \text{ mg N L}^{-1}$ (mean of $3,639 \text{ kg NANI km}^{-2} \text{ yr}^{-1}$) are lower than that in basin with lake N removal ratio $\leq 100\%$ and TN concentrations $> 1 \text{ mg N L}^{-1}$ (mean of $4,732 \text{ kg NANI km}^{-2} \text{ yr}^{-1}$). This suggests that whether a lake is in a self-purifying state depends on its N removal capacity and external N loading. Therefore, our study is the first to evaluate the relative relationship between external N

loading and lake N removal, and suggests that save resources by not focusing on lakes that already meet the water quality criteria and lakes are likely to reach this level with no management measures. Overall, our model provides a tool for assessing whether a watershed needs to reduce N input and quantifying the amount of required reduction of N input.

Response Fig. 4 Lake count and watershed net anthropogenic N inputs (kg NANI km⁻² yr⁻¹) among lake types with different N concentrations and N removal ratios.

Ln 136-139. And this basically means that if external N loading to lakes is high, their water quality tends to be worse and therefore external N loading should be reduced? Is there in fact a pattern that lakes with high ext N loading are more efficient in N removal than some of those with low ext N loading? That would be a more interesting finding.

[Response] Thank you for your insightful comment. We concluded that, in general, higher ext N loading resulted in lower N removal efficiencies, characterized by the biosaturation effect (Supplementary Fig. 7). However, as stated above, both the N concentration in the lake water and the biochemistry and environmental factors affect the activity of the denitrifying microbial process. Some studies have shown that N removals are more efficient in warmer, shallower, smaller, and more evenly mixed lakes^{5,51,55}. Due to the limitations of the data on lake properties and the focus of the manuscript, such cases are not discussed carefully. However, your insights have given us great inspiration, and we will continue to explore this issue in depth in future work.

Ln 180-181. compared with each preceding year or the initial situation before the reductions?

[Response] Compared with each preceding year. We have revised it accordingly throughout the revised manuscript. Thank you for helping us refine this expression better. (lines 182-183)

Ln 225-226. what's the unit for the water quality improvement efficiency here?

[Response] The water quality improvement efficiency is dimensionless and is the ratio of the reduction in lake N storage to the reduction in lake N loading. We have clarified this in the revised manuscript. (lines 246-248)

Ln 231-232. or: the lakes do not yet meet the water quality goal (exceed the goal sounds like they are already beyond the goal)

[Response] Thanks, revised accordingly. (lines 254)

Ln 238-240. what kind of traits characterize these lakes?

[Response] Thanks, lakes with stronger N removal ratios have higher water quality improvement efficiencies (Fig. 4b).

Fig. 4 Implementation of strategies to achieve the UN's SDG's clean water target before 2030.

References

1. Wu, Z. et al. Imbalance of global nutrient cycles exacerbated by the greater retention

- of phosphorus over nitrogen in lakes. *Nat. Geosci.* **15**, 464-468 (2022).
2. Loeks-Johnson, B. M. & Cotner, J. B. Upper Midwest lakes are supersaturated with N₂. *Proc. Natl Acad. Sci. USA* **117**, 17063 (2020).
 3. Zhang, W., Li, H. & Li, B. Whole-system estimation of hourly denitrification in a flow-through riverine wetland. *J. Hydrol.* **618**, 129132 (2023).
 4. Cheng, F. Y., Van Meter, K. J., Byrnes, D. K. & Basu, N. B. Maximizing US nitrate removal through wetland protection and restoration. *Nature* **588**, 625-630 (2020).
 5. Zhao, Y. et al. Nitrogen removal capacity of the river network in a high nitrogen loading region. *Environ. Sci. Technol.* **49**, 1427-1435 (2015).
 6. Li, X. et al. Sediment denitrification in waterways in a rice-paddy-dominated watershed in eastern China. *J. Soils Sediments* **13**, 783-792 (2013).
 7. Zhu, L. et al. Algal accumulation decreases sediment nitrogen removal by uncoupling nitrification-denitrification in shallow eutrophic lakes. *Environ. Sci. Technol.* **54**, 6194-6201 (2020).
 8. Yan, X. et al. A new model to estimate shallow lake nitrogen removal rate based on satellite derived variables. *Environ. Res. Lett.* **19**, 024025 (2024).
 9. Small, G. E., Cotner, J. B., Finlay, J. C., Stark, R. A. & Sterner, R. W. Nitrogen transformations at the sediment–water interface across redox gradients in the Laurentian Great Lakes. *Hydrobiologia* **731**, 95-108 (2014).
 10. Gardner, J. R., Fisher, T. R., Jordan, T. E. & Knee, K. L. Balancing watershed nitrogen budgets: accounting for biogenic gases in streams. *Biogeochemistry* **127**, 231-253 (2016).
 11. Zou, W. et al. Relationships between nutrient, chlorophyll *a* and Secchi depth in lakes of the Chinese Eastern Plains ecoregion: implications for eutrophication management. *J. Environ. Manag.* **260**, 109923 (2020).
 12. Qin, B. et al. Water depth underpins the relative roles and fates of nitrogen and phosphorus in lakes. *Environ. Sci. Technol.* **54**, 3191-3198 (2020).
 13. Seitzinger, S. P. et al. Nitrogen retention in rivers: model development and application to watersheds in the northeastern U.S.A. *Biogeochemistry* **57**, 199-237

- (2002).
14. Zhang, Y. et al. Dissolved oxygen stratification and response to thermal structure and long-term climate change in a large and deep subtropical reservoir (Lake Qiandaohu, China). *Water Res.* **75**, 249-258 (2015).
 15. Hanna, M. Evaluation of models predicting mixing depth. *Can. J. Fish. Aquat. Sci.* **47**, 940-947 (1990).
 16. Zhou, J., Leavitt, P. R., Zhang, Y. & Qin, B. Anthropogenic eutrophication of shallow lakes: Is it occasional? *Water Res.* **221**, 118728 (2022).
 17. Filazzola, A. et al. A database of chlorophyll and water chemistry in freshwater lakes. *Sci. Data* **7**, 310 (2020).
 18. Evans, R. D. Empirical evidence of the importance of sediment resuspension in lakes. *Hydrobiologia* **284**, 5-12 (1994).
 19. Smith-Morrill, L. The exchange of carbon, nitrogen, and phosphorus between the sediments and water-column of an amazon floodplain lake. *University of Maryland College Park*, (1987).
 20. Liu, C. et al. Nitrogen and phosphorus exchanges across the sediment–water interface in a bay of lake Chaohu. *Water Environ. Res.* **90**, 1956-1963 (2018).
 21. Pang, Y., Yan, R., Yu, Z., Li, Y. & Li, R. Suspension-sedimentation of Sediment and Release Amount of Internal Load in Lake Taihu Affected by Wind. *Environmental Science* **09**, 2456-2464 (2008).
 22. Dillon, P. J., Evans, R. D. & Molot, L. A. Retention and Resuspension of Phosphorus, Nitrogen, and Iron in a Central Ontario Lake. *Can. J. Fish. Aquat. Sci.* **47**, 1269-1274 (1990).
 23. Bloesch, J. Inshore–Offshore Sedimentation Differences Resulting from Resuspension in the Eastern Basin of Lake Erie. *Can. J. Fish. Aquat. Sci.* **39**, 748-759 (1982).
 24. Flower, R. J. Field calibration and performance of sediment traps in a eutrophic holomictic lake. *J. Paleolimnol.* **5**, 175-188 (1991).
 25. Rosa, F. Sedimentation and Sediment Resuspension in Lake Ontario. *J. Gt. Lakes*

- Res.* **11**, 13-25 (1985).
26. Kansanen, P. H., Jaakkola, T., Kulmala, S. & Suutarinen, R. Sedimentation and distribution of gamma-emitting radionuclides in bottom sediments of southern Lake Päijänne, Finland, after the Chernobyl accident. *Hydrobiologia* **222**, 121-140 (1991).
 27. Gasith, A. Tripton Sedimentation in eutrophic lakes—simple correction for the resuspended matter. *SIL Proceedings, 1922-2010* **19**, 116-122 (1975).
 28. Bachmann, R. W., Hoyer, M. V. & Canfield Jr, D. E. The Potential For Wave Disturbance in Shallow Florida Lakes. *Lake Reservoir Manag.* **16**, 281-291 (2000).
 29. Avnimelech, Y., Kochva, M. & Hargreaves, J. A. Sedimentation and Resuspension in Earthen Fish Ponds. *J. World Aquacult. Soc.* **30**, 401-409 (1999).
 30. Weyhenmeyer, G. A., Meili, M. & Pierson, D. C. A simple method to quantify sources of settling particles in lakes: Resuspension versus new sedimentation of material from planktonic production. *Mar. Freshw. Res.* **46**, 223-231 (1995).
 31. Qin, B. et al. Estimation of internal nutrient release in large shallow Lake Taihu, China. *Science in China Series D* **49**, 38-50 (2006).
 32. Zhong, J. et al. Nitrogen budget at sediment–water interface altered by sediment dredging and settling particles: Benefits and drawbacks in managing eutrophication. *J. Hazard. Mater.* **406**, 124691 (2021).
 33. Ni, Z. & Wang, S. Historical accumulation and environmental risk of nitrogen and phosphorus in sediments of Erhai Lake, Southwest China. *Ecol. Eng.* **79**, 42-53 (2015).
 34. Rocha, R. R. A., Thomaz, S. M., Carvalho, P. & Gomes, L. C. Modeling chlorophyll-a and dissolved oxygen concentration in tropical floodplain lakes. *Braz. J. Biol.* **69**, 491-500 (2009).
 35. Stefanidis, K. & Dimitriou, E. Differentiation in Aquatic Metabolism between Littoral Habitats with Floating-Leaved and Submerged Macrophyte Growth Forms in a Shallow Eutrophic Lake. *Water* **11**, 287 (2019).
 36. Tan, Z., Yao, H. & Zhuang, Q. A small temperate lake in the 21st century: dynamics of water temperature, ice phenology, dissolved oxygen, and Chlorophyll-a. *Water*

- Resour. Res.* **54**, 4681-4699 (2018).
37. Piña-Ochoa, E. & Álvarez-Cobelas, M. Denitrification in aquatic environments: a cross-system analysis. *Biogeochemistry* **81**, 111-130 (2006).
38. David, M. B., Wall, L. G., Royer, T. V. & Tank, J. L. Denitrification and the nitrogen budget of a reservoir in an agricultural landscape. *Ecol. Appl.* **16**, 2177-2190 (2006).
39. Bruesewitz, D. A., Hamilton, D. P. & Schipper, L. A. Denitrification potential in lake sediment increases across a gradient of catchment agriculture. *Ecosystems* **14**, 341-352 (2011).
40. Jiang, X. et al. Salinity-linked denitrification potential in endorheic Lake Bosten (China) and its sensitivity to climate change. *Front. Microbiol.* **13**, (2022).
41. Müller, B., Meyer, J. S. & Gächter, R. Denitrification and nitrogen burial in Swiss Lakes. *Environ. Sci. Technol.* **56**, 2794-2802 (2022).
42. Liu, W., Jiang, X., Zhang, Q., Li, F. & Liu, G. Has submerged vegetation loss altered sediment denitrification, N₂O production, and denitrifying microbial communities in subtropical lakes? *Global Biogeochem. Cy.* **32**, 1195-1207 (2018).
43. Palacin-Lizarbe, C., Camarero, L., Hallin, S., Jones, C. M. & Catalan, J. Denitrification rates in lake sediments of mountains affected by high atmospheric nitrogen deposition. *Sci. Rep.* **10**, 3003 (2020).
44. Zhong, J. et al. The co-regulation of nitrate and temperature on denitrification at the sediment-water interface in the algae-dominated ecosystem of Lake Taihu, China. *J. Soils Sediments* **20**, 2277-2288 (2020).
45. She, D. et al. Limited N removal by denitrification in agricultural drainage ditches in the Taihu Lake region of China. *J. Soils Sediments* **18**, 1110-1119 (2018).
46. Yao, X., Zhang, L., Zhang, Y., Xu, H. & Jiang, X. Denitrification occurring on suspended sediment in a large, shallow, subtropical lake (Poyang Lake, China). *Environ. Pollut.* **219**, 501-511 (2016).
47. Zhang, L. et al. Influence of long-term inundation and nutrient addition on denitrification in sandy wetland sediments from Poyang Lake, a large shallow subtropical lake in China. *Environ. Pollut.* **219**, 440-449 (2016).

48. Chen, N., Wu, J., Chen, Z., Lu, T. & Wang, L. Spatial-temporal variation of dissolved N₂ and denitrification in an agricultural river network, southeast China. *Agric. Ecosyst. Environ.* **189**, 1-10 (2014).
49. Liu, M. et al. Thermal stratification dynamics in a large and deep subtropical reservoir revealed by high-frequency buoy data. *Sci. Tot. Environ.* **651**, 614-624 (2019).
50. Small, G. E. et al. Large differences in potential denitrification and sediment microbial communities across the Laurentian great lakes. *Biogeochemistry* **128**, 353-368 (2016).
51. Cheng, F. Y. & Basu, N. B. Biogeochemical hotspots: Role of small water bodies in landscape nutrient processing. *Water Resour. Res.* **53**, 5038-5056 (2017).
52. Basu, N. B. et al. Managing nitrogen legacies to accelerate water quality improvement. *Nat. Geosci.* **15**, 97-105 (2022).
53. D'Arcy, B. & Frost, A. The role of best management practices in alleviating water quality problems associated with diffuse pollution. *Sci. Total Environ.* **265**, 359-367 (2001).
54. Chen, L., Wei, G. & Shen, Z. Incorporating water quality responses into the framework of best management practices optimization. *J. Hydrol.* **541**, 1363-1374 (2016).
55. Alexander, R. B., Smith, R. A. & Schwarz, G. E. Effect of stream channel size on the delivery of nitrogen to the Gulf of Mexico. *Nature* **403**, 758-761 (2000).
56. Jordan, S. J., Stoffer, J. & Nestlerode, J. A. Wetlands as sinks for reactive nitrogen at continental and global scales: a meta-analysis. *Ecosystems* **14**, 144-155 (2011).
57. Finlay, J. C., Small, G. E. & Sterner, R. W. Human influences on nitrogen removal in lakes. *Science* **342**, 247-250 (2013).
58. Hansen, A. T., Dolph, C. L., Fofoula-Georgiou, E. & Finlay, J. C. Contribution of wetlands to nitrate removal at the watershed scale. *Nat. Geosci.* **11**, 127-132 (2018).

Point-by-point Responses to Comments from the Reviewers

The authors would like to thank the efforts of the editorial office personnel and the reviewers for their comments and suggestions. Reviewer 3's remaining comments are responded to here, and we thank the reviewer again for helping us improve the focus of our work.

Responses to Reviewer #3's Comments

[Comments] *I kindly thank the authors for their responses to the review comments and suggestions and implementing these into the manuscript. In my opinion, several key points are now presented more clearly and concisely and I think the manuscript improved.*

What I would suggest the authors to still work on is the key message of this paper, which I still find somewhat ambiguous. The title or the paper, as well as the research questions stated in the introduction, suggest that the goals of this study were to show how the N removal efficiency of lakes is a key characteristic in defining their water quality – and thus their response to N loading reductions – and, by using the remote sensing model presented in this study, to identify the lakes in their dataset where N loading reductions lead to the most effective results. At the same time, the authors state:

“Our study provides an effective tool to estimate watershed scale N removal in lakes through remote sensing techniques.”

Is the main purpose of this paper to offer a new tool (i.e., the remote sensing model the authors developed and used for this study) to be used for any area/watershed/lake in order to determine where N loading should be targeted to yield best results in the lake water quality improvement – hence the key message of this paper being the introduction of this method as a tool for lake management? Or is the main purpose to show, by using this remote sensing model, that there is variability in the N removal efficiency of lakes globally, and that this should be taken into account while planning N reduction strategies?

Both these aspects are relevant per se, but they are different approaches to the theme – one is to introduce a new tool/methodology (the remote sensing model for N budget in lakes), and the other is to present a natural phenomenon (variability in N removal efficiency between lakes) by using the tool in question. Is the scope and target of the paper clear enough? I do not mean to question the methodology or results of the paper, but I was left with the impression that the authors in fact aimed to answer more research questions than were presented in the introduction, and hence I feel like the common thread is sometimes lost along the manuscript.

However, this could also be just my different thinking or perhaps a conceptual

misunderstanding, and if the Editor and other reviewers do not see any significant discontinuity in the manuscript, the authors can choose to ignore this feedback.

[Response] Thank you for your thoughtful and valuable comments.

We totally agree that the distinction between introducing a new tool and exploring the variability in N removal efficiency among lakes should be clearer. After careful consideration, we think that the primary aim of our study is to demonstrate, through the use of a remote sensing model, the significant global variability in N removal efficiency across lakes. This variability should be taken into account when planning watershed N input reduction strategies. The main contribution of our study is to present a natural phenomenon—the variability in N removal efficiency among lakes—and to propose a quantitative watershed N input reduction strategy to achieve the SDG of global clean water in lakes, using the remote sensing tool in question.

To address this concern, we have removed the sentence "Our study provides an effective tool to estimate watershed scale N removal in lakes through remote sensing techniques" to avoid any ambiguity. Additionally, we have clarified that we used the remote sensing model solely to answer the key question of this study—understanding global lake N removal efficiency and providing watershed nitrogen management options. To further enhance clarity, we have revised the Introduction to highlight one of the main challenges in understanding the relationship between watershed N management and lake water quality improvement—the difficulty in quantifying N removal from lakes. The revised sentence is as follows:

"This may be attributed to accurately quantifying the varied N removal rates in lakes through denitrification is challenging due to the high background concentrations of atmospheric N₂, especially at the landscape scale". (Ln 60–63)

We hope these revisions clarify the focus of our work.

Thank you again for your valuable insights. We believe these revisions enhance the clarity of our message.